# Daily Life Changes and Life Satisfaction among Korean School-Aged Children in the COVID-19 Pandemic

**DOI:** 10.3390/ijerph18063324

**Published:** 2021-03-23

**Authors:** Jihye Choi, Youjeong Park, Hye-Eun Kim, Jihyeok Song, Daeun Lee, Eunhye Lee, Hyeonjin Kang, Jeeho Lee, Jihyeon Park, Ji-Woo Lee, Seongeun Ye, Seul Lee, Sohee Ryu, Yeojeong Kim, Ye-Ri Kim, Yu-Jin Kim, Yuseon Lee

**Affiliations:** 1Department of Child Development and Family Studies, College of Human Ecology, Seoul National University, Seoul 08826, Korea; cjh14@snu.ac.kr (J.C.); hyeeun1996@snu.ac.kr (H.-E.K.); wkdrn1117@snu.ac.kr (J.S.); dlekdms2018@snu.ac.kr (D.L.); leh1288@snu.ac.kr (E.L.); huenjin2373@snu.ac.kr (H.K.); ckck0817@snu.ac.kr (J.L.); pjh1189@snu.ac.kr (J.P.); wooyou07@snu.ac.kr (J.-W.L.); yeseongeun97@snu.ac.kr (S.Y.); as50603@snu.ac.kr (S.L.); shryu916@snu.ac.kr (S.R.); yeoj0218@snu.ac.kr (Y.K.); yallavie21@snu.ac.kr (Y.-R.K.); you8972@snu.ac.kr (Y.-J.K.); cosmos1219@snu.ac.kr (Y.L.); 2Research Institution of Human Ecology, Seoul National University, Seoul 08826, Korea

**Keywords:** COVID-19 pandemic, life satisfaction, parent-child relationship, psychological well-being, school-aged children

## Abstract

The recent COVID-19 pandemic has been disrupting the daily lives of people across the world, causing a major concern for psychological well-being in children. This study aimed to examine (1) how life satisfaction and its potential predictors have been affected by the pandemic among school-aged children in Korea, and (2) which factors would predict their life satisfaction during the pandemic. We surveyed 166 fourth-graders in the Seoul metropolitan area to assess their psychological well-being and potentially related variables during the pandemic. The data were compared with those available from two pre-COVID-19 surveys, the 2018 Korean Children and Youth Panel Survey (*n* = 1236) and the 2019 Korean Children and Youth Well-being Index Survey (*n* = 334). Higher levels of stress were observed in children during the COVID-19 pandemic; however, the level of their life satisfaction remained unchanged when compared with data from the pre-COVID-19 surveys. The pandemic also affected peer relationship quality and susceptibility to smartphone addiction, but not perceived parenting style nor academic engagement. Interestingly, peer relationship quality no longer predicted life satisfaction during the pandemic; perceived parenting styles and parent-child conversation time predicted life satisfaction. The results suggest a central role of parent-child relationship in supporting the psychological well-being of school-aged children during the pandemic.

## 1. Introduction

Humans, when faced with risks, adapt to new environmental conditions by changing their lifestyles. The outbreak of the novel Coronavirus disease (COVID-19) pandemic led many countries to implement social distancing measures such as mass quarantine, which in turn affected the daily lives of people all over the world profoundly.

In South Korea, the government preemptively implemented strict social distancing measures from March 2020. As part of the measures, primary school classes were offered exclusively online at first. However, with growing concern about students’ learning loss due to the lack of teacher-child interaction, it was switched to a hybrid model in June 2020. Accordingly, children were required to physically go to school one to three days a week depending on school size, and attend school from home for the remaining weekdays. At school, children were asked to keep their masks on and minimize conversations with peers. They were also seated in a manner to ensure maximum distance between each other, further separated by plexiglass barriers. At home, children took classes by watching a series of pre-recorded or online lecture videos.

Given all the unusual demands imposed by such external environments, it is important to examine how children adapt to altered surroundings and the contributing factors. In the present study, we aimed to explore this by studying their daily lives and psychological well-being.

### 1.1. Children’s Lifestyles during the COVID-19 Pandemic

Major changes for school-aged children under this pandemic include adjusting to online classes (i.e., home-based learning), restriction of social gatherings and outdoor activities, and increased time spent home with parents. A considerable number of studies have reported that children’s lifestyles have transformed owing to these changes. One response to the online classes, where there is no commuting to school but more flexibility in schedules, is an increase in children’s sleep duration by approximately 0.5 to 0.65 h per day [1,2,3], coupled with a delayed bed time and wake time [1]. Although these findings importantly show how children’s sleep patterns have changed during the pandemic, they provide little information about whether the children’s sleep patterns vary with the type of school attendance (i.e., online versus face-to-face class days) during the COVID-19 pandemic. Korea provides a good arena for such comparison because it adopted a hybrid model of online and face-to-face classes. Comparing children’s time use on online versus face-to-face class days during the pandemic may capture one facet of children’s adaptation to the hybrid model of school attendance.

Another well-established change in daily life during the pandemic concerns children’s screen time. Several studies have reported a dramatic rise in screen consumption among children 5 to 18 years old [3,4,5,6]. For example, it has been reported that children’s screen time has increased by 4.85 h per day [5] and about 30 h per week [6] during the pandemic, raising concerns of digital technology overuse [7]. However, previous studies with primary school children examined the overall screen use that included online learning and leisure activities such as playing video games, watching TV, and communicating with friends and family, but did not focus on their smartphone use time separately from other types of screen time. Given that a majority (81.2%) of fourth-to sixth-grade children have smartphones [8] and that there is a growing concern about children’s smartphone overuse [9,10], investigating the changes in smartphone use among primary school children may provide a basis for designing interventions. We predicted that school-aged children’s smartphone use time would have been prolonged as, with restricted opportunities for other leisure activities, smartphones could be a means for communicating with friends and playing games. However, extended smartphone use might have affected children to depend more on smartphones, as shown in the case of increased internet addiction in adults within the pandemic [11]. The potential risk of smartphone addiction needs close attention as it may harm children’s psychological well-being [12] and academic performance [13].

In addition, the pandemic and consequentially, social distancing measures, have increased the time children spend with parents at home [14]. Yet, it is unclear whether parent-child conversation time has simultaneously increased. A study found that the negative impacts of COVID-19 on fourth and sixth graders’ psychological health, such as anxiety about the future and depressive symptoms, were larger in children whose parents were not with them during daytime [15]. Further, parent-child conversation has been shown to be the key to children’s life satisfaction during the pandemic [16]. Thus, investigating parent-child conversation time during the ongoing pandemic is important. We predicted that as parents and children stayed together for a longer duration during the pandemic, parent-child conversation time might have increased and played a protective role against negative environmental conditions of the child’s well-being.

### 1.2. Children’s Social Relationship during the COVID-19 Pandemic

Social support from parents and peers is a powerful protective factor for children’s positive development [17] and recovery from large-scale traumatic events [18,19,20]. However, there are concerns of heightened parenting stress and lowered parenting quality in the COVID-19 pandemic [21,22,23]. Spending more hours on childcare each day while managing their work remotely from home [24], along with other pandemic-related factors such as employment loss, financial insecurity, lack of help from external families, and a dearth of leisure time, may increase the risk of parental burnout [23]. Increased parenting stress can provoke negative parenting behaviors such as harsh parenting [21,25] and verbal hostility toward children [26]. Indeed, in a study, Korean school-aged children reported experiencing more conflicts, worries, and scolding from their parents during the pandemic [15].

Nevertheless, more empirical research is needed to understand the impact of the pandemic on parent-child relationship. Particularly, it is possible that parenting styles have not been detrimentally impacted despite pandemic-related challenges. First, parenting style may be considerably stable, as it is defined as the values and goals that parents have in socializing their children, the parenting practices they employ, and the attitudes they express toward their children [27]. Consistent with this possibility, some studies conducted in the United Kingdom and Korea found that most parents reported no change in parent–child relationship compared to the pre-COVID-19 period [28,29].

Another social relationship that plays a major role in children’s lives is peer relationships. Peer intimacy increases between the ages of 11 and 14 years [30]. Children’s perceived peer relationship and support have been shown to be linked to life satisfaction [17] and self-esteem [31]. However, the COVID-19 pandemic might interfere with peer relationship quality by reducing opportunities to interact with peers. Yet, to our knowledge, no studies have directly examined the influence of COVID-19 on peer relationship quality of primary school children.

Together, understanding whether COVID-19 has changed children’s relationships with parents and peers and whether they are still meaningful predictors of children’s life satisfaction during the pandemic could provide a basis for designing relation-based interventions for children’s psychological well-being in the COVID-19 era.

### 1.3. Children’s Academic Engagement during the COVID-19 Pandemic

Academic engagement, defined as a positive and fulfilling state of mind related to academic work [32], is an indicator of school adaptation of children [32,33] and a good predictor of life satisfaction of early youth [34,35]. School closures and online classes in the pandemic provoked concerns about the decrease in children’s academic engagement [36]. Online classes have led to restricted opportunities for teacher–child interaction, thus making it difficult for children to develop a secure and supportive relationship with their teachers [37], which is a known predictor of academic motivation for third to eighth grade students [31,38,39]. Additionally, children’s lack of learning interests has been reported by parents and teachers as one of the major difficulties in distance learning [40,41]. However, these findings were based on adult observations. To better understand whether there has been an actual change in academic engagement (which is a state of mind rather than behaviors related to academic work) since the outbreak of the pandemic, it is necessary to investigate children’s own report on their academic engagement.

### 1.4. Children’s Stress and Life Satisfaction during the COVID-19 Pandemic

Previous studies reported a rise in various stress responses and mental health issues during the COVID-19 pandemic [12,42,43,44], such as higher levels of anxiety symptoms and more prevalent clinical depression symptoms. Existing research, however, mainly focused on behavioral and emotional disorders from the mental health clinicians or pediatrics’ perspectives, resulting in limited evidence regarding children’s everyday stress level during the current pandemic. Understanding how the pandemic has affected children’s everyday stress would provide us with essential information about children’s current psychological well-being. Particularly, there is a need to closely examine the sub-areas of children’s daily stress, including family, academic performance, peer relationships, and financial pressure [45,46,47]. It is possible that stress in some contexts is exacerbated, while in other contexts, decreases or does not change under the pandemic. For example, stress related to home life might have increased since children spent more time at home, whereas stress related to peer pressure might have decreased as children had less chance to meet their peers. To our knowledge, this possibility has not been explored yet.

Life satisfaction has been a widely used indicator of psychological well-being in children [48]. Relatively stable, subjective well-being of a child is affected by accumulated experiences mainly in the long term rather than events in the short term [49,50]. Only a few studies have examined school-aged children’s life satisfaction in relation to the COVID-19 pandemic. In a study conducted in China in March 2020, children were generally found to be satisfied with life; 21.4% of the respondents reported that they had become more satisfied with life [16]. However, a study conducted in Australia in May 2020 found a significant decrease in life satisfaction among adolescents [51]. It is possible that children’s life satisfaction decreased as the pandemic prolonged, and varied with the specific quarantine measures implemented in different countries. There has been no investigation of life satisfaction in Korean school-aged children since the COVID-19 outbreak. Examining children’s life satisfaction in the midst of a pandemic (October 2020) would inform us of how profoundly children’s psychological well-being has been impacted by it. Thus, defining the predictors of children’s life satisfaction before the pandemic versus during it may provide clues to children’s adaptations to the novel environment.

### 1.5. Current Study

This study was designed to examine the differences in lifestyles, social relationships, academic engagement, and psychological well-being between pre- and during-COVID-19 pandemic in fourth-graders in Korea. We aimed to directly compare the same-grade children’s responses to the same questionnaire items, collected either before or amid the COVID-19 pandemic. To our knowledge, this study is among the first cross-cohort studies, as most prior studies used retrospective self-report after the pandemic, with a few studies employing longitudinal designs [3,51]. The cross-cohort data would uniquely contribute to understanding the impact of the pandemic, with direct statistical tests on the differences among same-aged cohorts, while controlling for potential age effects. The second goal of this study was to identify the predictors of life satisfaction in fourth-grade children before and during the COVID-19 pandemic. Thereby, we aimed to provide evidence that serves as a basis for designing interventions to support child psychological well-being during the COVID-19 pandemic.

## 2. Materials and Methods

### 2.1. Data and Participants

To examine changes in the daily lives and psychological well-being of fourth graders before and during the COVID-19 pandemic, we collected data in 2020 and compared them with data from two national surveys conducted in 2018 and 2019. First, a sample of the pre-COVID-19 fourth graders was obtained from the wave 1 data of the 2018 Korean Children and Youth Panel Survey (KCYPS), conducted by the National Youth Policy Institute. KCYPS is a nationwide longitudinal study that has been following up the development of 2607 children since 2018, when they were fourth-grade elementary school students. For the present study, we used the data of fourth graders living in the Seoul Metropolitan Area only (*n* = 1236). In addition, we used stress data from the 2019 Korean Children and Youth Well-being Index Survey (KCYWI), which has been administered annually to elementary school students of grade four to high school students across the country. We only used the data from fourth-grade elementary school children living in the Seoul Metropolitan Area who responded to all 11 questions about their stress level (*n* = 334).

For the during-COVID-19 group, a total of 168 fourth-graders (ranging from 9 to 10 years old) from the Seoul Metropolitan Area participated from 14 September to 5 October 2020. Two of them were excluded from the final sample because their responses were considered insincere (e.g., reporting an impossible amount of time for a day), yielding the final sample of 166 children (84 boys) in this group. We checked whether the sample size was appropriate to produce sufficient power, using G-Power version 3.1.9.4. Detailed protocols are available in Appendix A. Post hoc power analyses showed that, given the inputs (medium effect size *d* = 0.5, α = 0.05, sample size group 1 = 166, sample size group 2 = 334 and 1236), the independent *t* tests would likely detect existing effects (power = 0.9995079 and 0.9999779, respectively). In addition, a post hoc power analysis for a linear multiple regression model with a final sample size of 158 (8 invalid answers omitted), four predictors, an effect size of 0.15, and α = 0.05 indicated a power of 0.9823659.

### 2.2. Procedure and Measures

Most children responded to the survey by using a smartphone, a tablet, or a computer. Few of them, who were unable to respond online, responded using a paper survey. The survey assessed the following variables: daily time use, smartphone addiction proneness, academic engagement, peer relationship quality, perceived parenting style, stress, and life satisfaction.

#### 2.2.1. Daily Time Use

Children reported their wake time and bed time for weekdays and weekends. Since schools started online classes during the COVID-19 pandemic, we divided weekdays into online and face-to-face school days for the during-COVID-19 group. Sleep duration was calculated from the bed time and wake time. For the parent-child conversation time and smartphone use time on weekdays, the type of questions differed by cohort. Children in the during-COVID-19 group answered the time durations in minutes, whereas those in the pre-COVID-19 group answered the categorical questions.

#### 2.2.2. Smartphone Addiction Proneness

The Smartphone Addiction Proneness Scale [52] is a 15-item measure that captures the extent to which students psychologically rely on smartphone use in their daily life using a four-point Likert scale. The scale assesses the following sub-factors: disturbance of adaptive functions, virtual life orientation, withdrawal, and tolerance. Sample items include *My school grades dropped due to excessive smartphone use* (disturbance of adaptive functions); *Using a smartphone is more enjoyable than spending time with family or friends* (virtual life orientation); *It would be painful if I am not allowed to use a smartphone* (withdrawal); and *Even when I think I should stop, I continue to use my smartphone too much* (tolerance). Higher scores indicate increased smartphone addiction proneness. In the present study, Cronbach’s α, a common estimate of internal consistency of a multiple-item scale [53], was used to check the reliability of the scale. The reliability coefficient was Cronbach’s α = 0.814.

#### 2.2.3. Perceived Parenting Style

Children’s perception of parenting style was assessed with a 24-question scale, Parents as Social Context Questionnaire for Korean Adolescents (PSCQ_KA) [54], which was the Korean version of the original PSCQ [55]. The questionnaire contained six sub-factors: three positive aspects of parenting style (warmth, structure, and autonomy support) and three negative aspects of parenting style (rejection, chaos, and coercion). Children reported on a four-point Likert scale. Answers to items of negative parenting styles were reverse-coded, such as *Sometimes I wonder if my parents like me* (rejection). Cronbach’s α was 0.899 in the current sample.

#### 2.2.4. Peer Relationship Quality

We used the Korean Peer Relationship Quality Scale for Adolescents [56] for children’s perception of their peer relationship quality. The KPRQSA measures positive and negative peer relationships: intimate exchange, social support, satisfaction with peer relationships, conflict and opposition, one-sided leadership, withdrawal, and isolation. It consists of 13 items (e.g., *I spend time with my friends*; *I do not try to get along with others; I have good relationships with friends;* and *I have frequent disagreements with friends*) rated on a four-point Likert scale. Cronbach’s α was 0.801.

#### 2.2.5. Academic Engagement

Children’s academic engagement was measured using the Korean version of the Academic Engagement Inventory [57], consisting of 16 items, rated on a four-point Likert scale, about dedication, vigor, efficacy, and absorption with regard to studying. Dedication measures one’s understanding of the importance, value, and challenges of studying. Vigor focuses on the energy for studying consistently and cheerfully. Efficacy refers to confidence in studying, and absorption measures whether one fully concentrates on studying. Items include *I am capable enough to solve difficult tasks* and *I am confident in studying*. The Cronbach’s α for this scale was 0.912.

#### 2.2.6. Stress

Stress was assessed using the stress scale used by Y. M. Song and Y. J. Lee [58], which was a shortened version of the Stress Scale developed by J. H. Kim and D. W. Lee [59]. It consists of 12 items rated on a five-point scale and captures five sub-factors of adolescents’ everyday stress: family (e.g., *I am stressed because my parents interfere too much*), academic performance (e.g., *I am stressed because of homework or exams*), peer pressure (e.g., *I am stressed because my friends do not endorse me*), and financial pressure (e.g., *I am stressed because of insufficient pocket money*). We omitted a question related to academic performance that was not suitable for elementary school students (e.g., *I am stressed because of the college entrance exam or job finding*). This resulted in 11 items in total. Cronbach’s α was 0.817.

#### 2.2.7. Life Satisfaction

Children’s life satisfaction was measured with a Korean version of the Satisfaction with Life Scale (SWLS) developed by Diener, Emmons, Larsen, and Griffin [60]. It consists of five items, including, *In general, my life is close to my ideal*, and *The circumstances of my life are very good*. Measurements are made on a four-point Likert scale ranging from *I strongly disagree* (1 point) to *I strongly agree* (4 points). Cronbach’s α in the present sample was 0.815.

### 2.3. Data Analysis

Data were analyzed using SPSS 25.0 (IBM Corp., Armonk, NY, USA) [61]. First, descriptive statistics, independent samples *t* tests, and chi-square tests were used to investigate the presence of differences in daily lives between the pre-COVID-19 and during-COVID-19 fourth graders. Specifically, Welch’s *t* test [62], where degrees of freedom are estimated by the Welch–Satterthwaite equation, was used for protection against Type I error in unequal sample sizes [63]. In addition, paired samples *t* tests were employed to compare time spent between face-to-face and online school days among the during-COVID-19 fourth graders. Thereafter, Pearson’s correlation tests and multiple regression analyses were conducted to examine the variables predicting life satisfaction for the pre- and during-COVID-19 groups, respectively. Additionally, an ANCOVA was performed to further understand the relationship between peer relationship quality and life satisfaction in the pre- and during-COVID-19 cohorts.

## 3. Results

Table 1 presents the means and standard deviations for the variables that were measured in both the pre- and during-COVID-19 groups.

### 3.1. Daily Life Changes after the Outbreak of COVID-19

Our first set of analyses compared the daily lifestyle (sleep patterns, smartphone use time, smartphone addiction proneness, and parent-child conversation), social relationship (perceived parenting style and peer relationship quality), academic engagement, and psychological well-being (stress and life satisfaction) between the pre- and during-COVID-19 pandemic fourth-graders.

#### 3.1.1. Sleep Patterns

As shown in Table 1, children’s wake time, bed time, and sleep duration on face-to-face school days and weekends did not significantly differ between the pre- and during-COVID-19 groups. Children’s sleep patterns on online school days were not compared because there was no online school day before the COVID-19 pandemic.

#### 3.1.2. Smartphone Use Time

Smartphone use time was divided into three subgroups: <1 h, 1–3 h, and ≥3 h. Significant differences were observed for daily smartphone use time between the pre- and during-COVID-19 groups, χ^2^ (2) = 22.470, *p* < 0.001. The percentage of children who used smartphones for less than 1 h per day decreased from 56.6% to 39.5%, while the percentage of those who used smartphones 1 to 3 h per day increased from 35.0% to 42.7%, and the proportion of more-than-3-h smartphone users increased from 8.5% to 17.8%.

#### 3.1.3. Smartphone Addiction Proneness

The level of smartphone addiction proneness was significantly higher in the during-COVID-19 group (*M* = 1.97, *SD* = 0.61) than in the pre-COVID-19 group (*M* = 1.80, *SD* = 0.50), *t* (153) = −3.130, *p* = 0.002.

#### 3.1.4. Parent-Child Conversation Time

Parent-child conversation time was also classified into three subgroups: <1 h, 1–3 h, and ≥3 h. Daily time spent for parent-child conversation differed between the pre- and during-COVID-19 groups, χ^2^ (2) = 20.827, *p* < 0.001. A larger percentage of children had more than three hours of conversation with their parents per day in the during-COVID-19 group (31.6%) than the pre-COVID-19 group (18.2%), with a decrease in the percentage of children who had less than 1 h of conversation from 43.9% to 28.5%.

#### 3.1.5. Perceived Parenting Style and Peer Relationship Quality

The perceived parenting style showed no significant group differences. However, children’s perceived peer relationship quality was significantly lower in the during-COVID-19 group (*M* = 2.95, *SD* = 0.44) than in the pre-COVID-19 group (*M* = 3.05, *SD* = 0.42), *t* (197) = 2.755, *p* = 0.006.

#### 3.1.6. Academic Engagement

Academic engagement did not significantly differ between the pre- and during-COVID-19 groups.

#### 3.1.7. Stress

The children’s overall stress level was significantly higher in the during-COVID-19 group (*M* = 2.03, *SD* = 0.72) than in the pre-COVID-19 group (*M* = 1.72, *SD* = 0.78), *t* (311) = −4.276, *p* < 0.001. An examination of the sub-factors revealed a significant increase in the stress related to family (*p* < 0.001), academic performance (*p* < 0.001), and financial pressure (*p* < 0.05), but not in the stress related to peer pressure.

#### 3.1.8. Life Satisfaction

Life satisfaction did not significantly differ between the pre- (*M* = 3.11, *SD* = 0.57) and during-COVID-19 (*M* = 3.10, *SD* = 0.66) groups.

Thus, compared to children of the same age before the COVID-19 pandemic, the fourth-graders in the COVID-19 pandemic appeared to spend more time using smartphones and conversing with parents. They also showed more smartphone addiction proneness, higher levels of stress, and lower levels of peer relationship quality. In contrast, children’s sleep patterns of face-to-face school days and weekends, perceived parenting style, academic engagement, and life satisfaction did not show any changes.

#### 3.1.9. Additional Analyses on Differences in Daily Time Use between Online and Face-to-Face School Days.

We performed an additional analysis to determine whether the during-COVID-19 fourth graders used their time differently depending on whether it was a face-to-face school day or an online school day. As shown in Table 2, the during-COVID-19 fourth-graders woke up later (8:07 vs. 7:32, *p* < 0.001), went to bed later (22:41 vs. 22:18, *p* < 0.001), and spent more time sleeping (9:26 vs. 9:14, *p* = 0.001) on online school days than they did on face-to-face school days. In addition, the daily duration of parent-child conversation (146 min vs. 115 min) and child’s smartphone use (101 min vs. 80 min) were significantly longer on online school days than those on face-to-face school days (*p*s < 0.001). Together, the results indicate that the daily time use of fourth-grade children varies with the type of school attendance during the COVID-19 period.

### 3.2. Predictors of Life Satisfaction before and during the COVID-19 Pandemic

In order to identify the predictors of children’s life satisfaction before and during the COVID-19 pandemic, we first calculated Pearson’s correlations between the variables in each group. As can be seen in Table 3, life satisfaction was positively correlated with perceived parenting style, *r* = 0.51, *p* < 0.001, peer relationship quality, *r* = 0.40, *p* < 0.001, and academic engagement, *r* = 0.45, *p* < 0.001, in the pre-COVID-19 group. In the during-COVID-19 group, life satisfaction was positively related to perceived parenting style, *r* = 0.59, *p* < 0.001, peer relationship quality, *r* = 0.29, *p* < 0.001, academic engagement, *r* = 0.39, *p* < 0.001, and parent-child conversation time, *r* = 0.22, *p* < 0.01. These results confirmed that the four variables were adequate to be included in the regression as explanatory variables.

The regression for the pre-COVID-19 group was conducted using the enter method, in which all three independent variables were entered at the same time. However, the regression for the during-COVID-19 group was computed using the hierarchical method, where the three independent variables shared with the pre-COVID-19 group were entered in Model 1, and then one unique independent variable, parent-child conversation time, was added in Model 2. Linear regression for all three models (pre-COVID-19; Models 1 & 2 of during-COVID-19) was shown to be suitable by the residuals versus fitted values plots depicting low levels of heteroscedasticity. Table 4 displays the results of the regression analyses.

The regression model for the pre-COVID-19 group indicated an explained variance of 37.4%. Perceived parenting style (*B* = 0.514, *p* < 0.001), peer relationship quality (*B* = 0.206, *p* < 0.001), and academic engagement (*B* = 0.311, *p* < 0.001) significantly contributed to explaining the life satisfaction of the pre-COVID-19 fourth graders (all *p*s < 0.001). That is, a more positive perception of parents’ parenting style, better quality of peer relationships, and higher academic engagement were associated with higher levels of life satisfaction. The significance of coefficients remained when the child’s gender was controlled for. Among the significant variables, perceived parenting style (*β* = 0.362) had the largest effect on a child’s life satisfaction, followed by academic engagement (*β* = 0.281) and peer relationship quality (*β* = 0.153).

The regression for the during-COVID-19 group indicated an explained variance of 36.0% for Model 1 and 37.8% for Model 2, with 1.8% of the variance additionally explained by parent-child conversation time on face-to-face school days. The results from Model 1 indicated that perceived parenting style (*p* < 0.001) and academic engagement (*p* < 0.05) predicted life satisfaction as they did for the pre-COVID-19 group. However, peer relationship quality was no longer a significant predictor. When the parent-child conversation time was added to Model 2, it was found that it was a significant predictor of life satisfaction in the during-COVID-19 group (*p* < 0.05). Thus, particularly for the during-COVID-19 group, the longer the children had conversations with their parents on face-to-face school days, the more satisfied they felt with their lives. Again, controlling for the child’s gender, the significance remained. Additionally, among the significant predictors, perceived parenting style (*β* = 0.504) had the largest effect on the variance of child life satisfaction, followed by academic engagement (*β* = 0.185) and parent-child conversation time (*β* = 0.136).

To further understand the current finding that peer relationship quality no longer predicted life satisfaction in the during-COVID-19 group, each cohort of fourth graders was divided into four subgroups based on their scores on peer relationship quality (i.e., low, lower-middle, upper-middle, and high). Table 5 and Figure 1 display the estimated average values of life satisfaction by cohorts and peer relationship quality groups. As shown in Figure 1, the pre-COVID-19 group indicates a generally upward trend, with higher-quality peer relationship group showing higher life satisfaction, whereas the during-COVID-19 fourth graders show an inverted U-shaped trend, suggesting relatively low life satisfaction in the high quality peer relationship group. A 2 (upper-middle vs. high-quality peer relationship) × 2 (pre- vs. during-COVID-19) ANCOVA with perceived parenting style and academic engagement as covariates confirmed that this interaction was significant, *F* (1, 778) = 4.663, *p* = 0.031, *η*^2^ = 0.006, as presented in Table 6. Follow-up independent samples *t* tests indicated that the high quality peer relationship group was higher in life satisfaction than the upper-middle quality peer relationship group in the pre-COVID-19 group, *t* (702) = −7.218, *p* < 0.001, while the high quality peer relationship group had no difference in life satisfaction with the upper-middle quality peer relationship group in the during-COVID-19 group.

## 4. Discussion

Given the changes in children’s surroundings, the present study aimed to (1) examine the impact of the COVID-19 pandemic on the daily life and psychological well-being of school-aged children and (2) explore the predictors of life satisfaction in school-aged children before and during the COVID-19 pandemic.

Our findings reveal that the COVID-19 pandemic has influenced several aspects of daily life among school-aged children. We found evidence that primary school children’s smartphone use time and smartphone addiction proneness increased compared to the pre-COVID-19 period, as expected based on the previous findings in older age groups that smartphone use time increased during the COVID-19 pandemic [65,66]. Furthermore, our results showed that peer relationship quality has been damaged during the pandemic. This finding is likely attributed to the decreased opportunities to interact with peers during the pandemic (e.g., conversation restriction at school, and decreased playtime with friends). Considering that playtime with peers was positively correlated with peer attachment [67], decreased playtime with friends due to forced social distance and school closures would have negatively affected the quality of peer relationships. The results, though, also revealed that children had more time to have conversations with their parents during the pandemic. As children stayed longer with parents at home [14], it seems that the interaction with family members has been enlarged in quantity.

However, the present findings also elucidate some facets of daily life that have been unchanged by the pandemic. First, this study provides the first evidence that sleep schedules on face-to-face school days and weekends have not changed, although children reported delayed sleep schedules and longer sleep duration on online school days compared to face-to-face school days. This finding confirms the delayed sleep schedules and increased sleep duration reported in many studies [1,2,3] related to the shift to online classes. It also demonstrates that children’s sleep schedules vary according to online versus face-to-face classes. Second, academic engagement did not change during the ongoing pandemic. Although online classes disturbed the teacher–child interaction [37], several factors might have helped them to retain their vigor, efficacy, dedication, and absorption in studying. One possible external factor is parental involvement in home learning. Similar to the previous finding, that most parents were engaged in their children’s online learning activities [68,69], parents might have been sharing a greater burden of motivating children towards academic work with teachers during this pandemic. Another external protector might be the presence of face-to-face classes. The implementation of the hybrid model allowed interpersonal contact between teachers and children at least once a week. Thus, the outcome might differ in prolonged, complete school closure. Another novel finding of the current study is that perceived parenting style did not differ between before and during the COVID-19 pandemic. This finding may seem inconsistent with many findings, suggesting an increase in negative parenting behaviors [15,21,22,23,25,26]. However, this is in line with a few findings [28,29], supporting the possibility that parenting styles may be considerably stable and not easily affected by temporary stressful events or changes in external circumstances. However, it is also possible that the during-COVID-19 sample in the current study was biased such that the sample more likely consisted of children having positive parent-child relationships than children from at-risk families, because children’s participation required parental consent. Results may vary for studies with a larger sample that includes diverse at-risk families.

Regarding children’s psychological well-being, their overall stress level increased compared to the pre-COVID-19 period. This result is in line with recent studies that reflected that children’s worsened stress level and mental health are related to the pandemic [42,43,44]. As expected, the sub-areas of children’s stress varied in significance of change. While stress from family, school performance, and financial pressure increased, stress from peer pressure remained unchanged. Notably, social isolation could have triggered a higher level of stress [70,71], inducing more conflicts with family members, distracting schoolwork, and bringing about economic strain for households. Life satisfaction, however, remained unaffected. This result is consistent with the finding of a study where the life satisfaction score for the current state was generally high and the perceived change in life satisfaction was low in Chinese students [16]. The seemingly contradictory evidence of decreased level of life satisfaction in Australian adolescents [51] can be understood as stemming from age differences. Unlike primary school-aged children, secondary school-aged adolescents experience a drop in life satisfaction. The synthesis of the heightened stress and unchanged life satisfaction suggests the co-existence of negative impact and resilience in children during the pandemic, and affirms findings from the previous research [16].

Predictive factors for life satisfaction during the COVID-19 pandemic were found in all three aspects of life—daily time use, social relationship, and academic engagement. Children who had more conversations with their parents had higher subjective well-being levels. Together with the link between positive interactions among family members and children’s happiness [72], the present study suggests that increased positive interaction time with parents can contribute to children’s life satisfaction. Further, academic engagement affected children’s life satisfaction both before and during the COVID-19. This confirms previous findings that students’ engagement in academic activities positively relates to the life satisfaction level [73,74].

Perceived parenting style predicted children’s life satisfaction in the pre- and during-COVID-19 era. This result is consistent with previous research emphasizing the close relationship between the positive perceived parenting style and a higher level of happiness [75]. During the changes in children’s daily routines due to the pandemic situation, perceived parenting style did not change before and during the COVID-19 pandemic [76]. This can also work as a protective factor for children’s positive subjective well-being. Most importantly, peer relationship quality significantly impacted life satisfaction before the COVID-19 pandemic, but not in the current COVID-19 period. According to our further analysis, although the high-quality peer relationship group had higher life satisfaction in the pre-COVID-19 group, in the during-COVID-19 group, it resulted in the same level of life satisfaction with the upper-middle quality peer relationship group. This result suggests that the impact of the pandemic on psychological well-being might have been larger for children who had had a strong bond with peers and had spent a considerable time playing with their peers before the pandemic.

There are several limitations to the present study. First, the participants were limited to fourth graders, who were mostly 9- and 10-years-old. Thus, there may be limitations in representing school-aged children. Additionally, the data was collected in 2020, before the COVID-19 resurgence in South Korea. The longer the COVID-19 pandemic situation persists, the variables that did not differ between the pre-and during-COVID-19 groups may differ in the long term. Longitudinal studies can be conducted to better understand the prolonged impact of the COVID-19 pandemic on children’s daily lives and well-being. Moreover, we could not control socio-economic variables that might influence life satisfaction among children (e.g., parents’ education level) because we did not collect such data. It is an open question whether our finding that parent-child conversation time predicted life satisfaction would hold if the parents’ education level were controlled for. However, we believe that the current findings serve as a stepping stone for further research on predictors of children’s psychological well-being in the pandemic.

## 5. Conclusions

Despite the limitations, to our knowledge, this is one of the few studies to examine and compare the daily life of children before and during the COVID-19 pandemic. While most of the prior findings tend to highlight the negative consequences of the COVID-19 pandemic such as impaired daily routines, the present findings elucidate both what has been changed during the pandemic and what has not despite the social distancing. Furthermore, our findings emphasize a central role of parent-child relationship in supporting the psychological well-being of school-aged children during the pandemic. Some implications for practice can be drawn from the present findings. First, as the importance of parental roles has become even larger, parental education is needed to guide parents to have enough conversation time with their children and to monitor their smartphone use time to mitigate the negative effects on psychological well-being. Second, an increase in smartphone addiction proneness and decrease in peer relationship quality call for the intervention to prevent excessive smartphone use in playing games alone and to recover connection with other people. Meanwhile, considering the change in relation between peer relationship quality and life satisfaction after the pandemic, it is possible that children with high peer relationships are more vulnerable to its impact. Children might need different interventions based on their peer relationship levels. Providing technological devices for virtual interaction, such as video chats, might be helpful as suggested by researchers [77,78].

## Figures and Tables

**Figure 1 ijerph-18-03324-f001:**
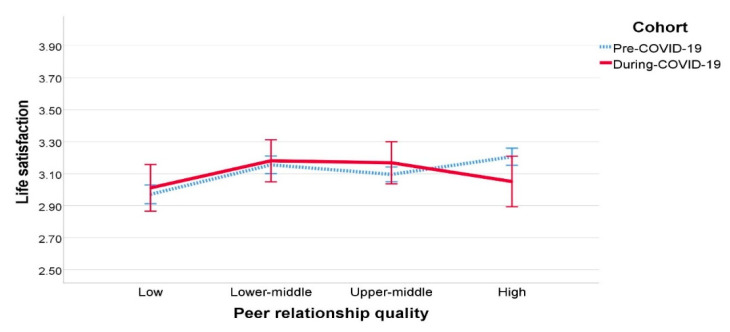
Estimated average values of life satisfaction by cohorts and peer relationship quality. Note. Covariates: perceived parenting style = 3.3083, academic engagement = 2.7203; bars represent 95% confidence intervals.

**Table 1 ijerph-18-03324-t001:** Descriptive statistics and group differences for daily life and psychological well-being variables.

	Pre-COVID-19 Pandemic (*n* = 1236)		During-COVID-19 Pandemic (*n* = 166)			
Variables	*M* or Frequency	*SD* or %	*M* or frequency	*SD* or %	*t* or χ^2^	*p* ^4^
Sleep patterns						
Face-to-face school day						
Wake time	7:36	0:27	7:32	0:28	1.733	0.085
Bed time	22:25	0:48	22:18	0:57	1.420	0.157
Sleep duration	9:11	0:48	9:14	0:57	−0.533	0.594
Weekend						
Wake time	8:37	1:10	8:46	1:13	−1.550	0.123
Bed time	22:59	0:57	23:02	1:06	−0.680	0.497
Sleep duration	9:38	1:14	9:44	1:15	−0.915	0.361
Smartphone use time ^1^						
<1 h	699	56.6%	62	39.5%	22.470	<0.001 ***
1–3 h	432	35.0%	67	42.7%		
≥3 h	105	8.5%	28	17.8%		
Smartphone addiction proneness ^2^	1.80	0.50	1.97	0.61	−3.130	0.002 **
Parent-child conversation time ^1^						
<1 h	542	43.9%	45	28.5%	20.827	<0.001 ***
1–3 h	469	37.9%	63	39.9%		
≥3 h	225	18.2%	50	31.6%		
Perceived parenting style	3.31	0.40	3.32	0.44	−0.259	0.796
Peer relationship quality	3.05	0.42	2.95	0.44	2.755	0.006 **
Academic engagement	2.73	0.51	2.68	0.49	1.229	0.220
Stress ^3^						
Overall	1.72	0.78	2.03	0.72	−4.276	<0.001 ***
Family	1.73	0.95	2.31	1.10	−5.608	<0.001 ***
Academic performance	1.92	1.03	2.28	0.98	−3.618	<0.001 ***
Peer pressure	1.71	1.04	1.72	0.89	−0.093	0.926
Financial pressure	1.51	0.82	1.70	0.79	−2.517	0.012 *
Life satisfaction	3.11	0.57	3.10	0.66	0.184	0.854

^1^ Average time spent per day on face-to-face school days. Note that time use variables of during-COVID-19 group were collected as continuous variables and converted to discrete ones. Ambiguous answers (e.g., “a lot of time”) and outliers (more than 1.5 times of interquartile range away from the 1st or 3rd quartile), as many as 13 responses, were not included in the analysis. Degrees of freedom was 2 for χ^2^; ^2^ We did not ask the participants who did not use smartphones about smartphone addiction proneness (108 participants of the pre-COVID-19 group, 12 participants of the during-COVID-19 group). We also excluded 21 participants for incomplete responses in the during-COVID-19 group; ^3^ Stress data of the pre-COVID-19 group were from 2019 Korean Children and Youth Well-being Index Survey (*n* = 334), unlike all the other variables of the pre-COVID-19 group were from KCYPS 2018 (*n* = 1236). We omitted 16 participants from stress data of the during-COVID-19 group due to incomplete responses; ^4^ Since multiple tests were performed on the same set of samples, Benjamini and Hochberg’s technique [64] was used to control False Discovery Rate. The adjusted *p*-values indicated that significance remained the same; * *p* < 0.05; ** *p* < 0.01; *** *p* < 0.001.

**Table 2 ijerph-18-03324-t002:** Descriptive statistics for daily time use of the during-COVID-19 fourth graders on face-to-face and online school days.

	During-COVID-19 (*n* = 166)					
	Face-to-Face School Day		Online School Day			
	*M*	*SD*	*M*	*SD*	*t*	*p* ^2^
Wake time	7:32	0:28	8:07	0:47	−11.868	<0.001 ***
Bed time	22:18	0:57	22:41	1:04	−7.805	<0.001 ***
Sleep duration (h)	9:14	0:57	9:26	0:57	−3.478	0.001 **
Smartphone use time (min) ^1^	80	70	101	83	−4.778	<0.001 ***
Parent-child conversation time (min) ^1^	115	93	146	112	−5.156	<0.001 ***

^1^ Ambiguous answers (e.g., “a lot of time”) and outliers (more than 1.5 times of interquartile range away from the 1st or 3rd quartile), as many as 13 responses, were not included in the analysis; ^2^ Since multiple tests were performed on the same set of sample, Benjamini and Hochberg’s technique [64] was used to control False Discovery Rate. The adjusted *p*-values indicated that significance remained the same; ** *p* < 0.01; *** *p* < 0.001.

**Table 3 ijerph-18-03324-t003:** Pearson correlations of perceived parenting style, peer relationship quality, academic engagement, parent-child conversation time, and life satisfaction.

	Pre-COVID-19 (*n* = 1236)			During-COVID-19 (*n* = 158)			
	2.	3.	4.	2.	3.	4.	5.
1. Life satisfaction	0.51 ***	0.40 ***	0.45 ***	0.58 ***	0.26 ***	0.36 ***	0.20 **
2. Perceived parenting style		0.41 ***	0.32 ***		0.40 ***	0.34 ***	0.12
3. Peer relationship quality			0.36 ***			0.39 ***	0.09
4. Academic engagement							0.05
5. Parent-child conversation time ^1^							

^1^ Parent-child conversation time on the face-to-face school day, modified by square root transformation to fix positive skewness. Note that parent-child conversation time was added to the analysis only for the during-COVID-19 group as it had not been measured in the pre-COVID-19 group; ** *p* < 0.01; *** *p* < 0.001.

**Table 4 ijerph-18-03324-t004:** Predictors of life satisfaction before and during COVID-19 pandemic.

	Pre-COVID-19			During-COVID-19					
				Model 1			Model 2		
Variables	*B*	*SE*	*β*	*B*	*SE*	*β*	*B*	*SE*	*β*
Perceived parenting style	0.514 ***	0.036	0.362	0.767 ***	0.106	0.517	0.748 ***	0.106	0.504
Peer relationship quality	0.206 ***	0.034	0.153	−0.016	0.111	−0.010	−0.026	0.110	−0.017
Academic engagement	0.311 ***	0.027	0.281	0.251 *	0.097	0.185	0.251 *	0.096	0.185
Parent-child conversation time ^1^							0.020 *	0.009	0.136
Intercept	−0.065	0.122		− 0.074	0.374		−0.174	0.373	
Δ R^2^							0.018		
Total R^2^	0.374			0.360			0.378		
*F*	245.443 ***			28.890 ***			23.267 ***		
Number of Observations	1236			158					

^1^ Parent-child conversation time on the face-to-face school day, modified by square root transformation to fix positive skewness; * *p* < 0.05; *** *p* < 0.001.

**Table 5 ijerph-18-03324-t005:** Estimated average values of life satisfaction by cohorts and peer relationship quality groups.

	Peer Relationship Quality Groups							
	Low		Lower-Middle		Upper-Middle		High	
	*M*	*SE*	*M*	*SE*	*M*	*SE*	*M*	*SE*
Cohorts								
Pre-COVID-19	2.97	0.03	3.16	0.03	3.10	0.02	3.21	0.03
During-COVID-19	3.01	0.07	3.18	0.07	3.17	0.07	3.05	0.08

Note. Covariates: perceived parenting style = 3.3083, academic engagement = 2.7203.

**Table 6 ijerph-18-03324-t006:** Two-way ANCOVA for life satisfaction by cohorts and peer relationship quality groups (upper-middle and high).

Resources	SS	Df	MS	*F*
Covariance (perceived parenting style)	20.418	1	20.418	105.926 ***
Covariance (academic engagement)	19.552	1	19.552	101.437 ***
Cohorts	0.101	1	0.101	0.526
Peer relationship quality groups	0.001	1	0.001	0.009
Cohorts x Peer relationship quality groups	0.899	1	0.899	4.663 *

Note. SS = Sum of Squares; Df = Degree of freedom; MS = Mean Square; * *p* < 0.05; *** *p* < 0.001.

## Data Availability

Publicly available datasets were analyzed in this study. The data of 2018 Korean Children and Youth Panel Survey (KCYPS) can be found here: https://www.nypi.re.kr/archive/mps. Data of the 2019 Korean Children and Youth Well-being Index Survey (KCYWI) are openly available in KOSSDA at http://hdl.handle.net/20.500.12236/23561, reference number A1-2019-0001. The data generated in this study are available on request from the corresponding author. The data are not publicly available due to privacy.

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
