# Peer review of "Daily Life Changes and Life Satisfaction among Korean School-Aged Children in the COVID-19 Pandemic"

_ijerph, 2021, doi:10.3390/ijerph18063324_

Round 1
Reviewer 1 Report
This paper undertakes a first-in-kind cross-cohort study to compare psychological well-being of school children prior to and during the COVID-19 pandemic in Korea. The paper is extremely well written, showcasing interesting results and benefitting from a very clear presentation and an exhaustive literature review. Overall, it is a pleasure to read. The results found by the authors are sound and provide useful insights by identifying significant drivers in children's response to the pandemic and the resulting changes in lifestyles. The discussion is interesting, and challenging some earlier results, for example, in relation to parenting behaviors. I am happy to recommend the paper for publication subject to a satisfactory revision to address the following points:
- Line 253 and throughout: "The Cronbach’s α" — change to "The Cronbach α" or "Cronbach’s α".
- Before using Cronbach's α, it is worth reminding briefly of the meaning and purpose of this coefficient, perhaps adding a suitable background reference.
- Line 301: change "ask ... to participants" to "ask ... the participants" (e.g., replace "to" with "the").
- Lines 310, 311"n=... ": use $n=...$ to produce math italics, n=....
- Table 2, heading (page 9): change "N=166" to "n=166" (cf. Table 1, heading of the right column).
- Lines 325-331 (smartphone use time): perhaps you can comment that there are three bins (<1h, 1-3h, and ≥3h), hence you use χ2 with 3-1=2 degrees of freedom. Same for the parent-child conversation time (lines 336-341).
- Line 338: Change χ2 to χ2 (cf. line 327).
- Lines 332-335 and the corresponding row in Table 2: please say more about how you use the t-test, including how the degrees of freedom are calculated. This is an approximate t-test, because the variances in the two groups are possibly different. Of course, you are using SPSS for testing, but it is useful to make sense of what is going on. I have attempted to calculate the t-statistic but the results are slightly different from what you have reported. Specifically, I used the statistic T = (M1-M2)/√(SD1/n1+SD2/n2), which has approximated by t-distribution with degrees of freedom k = min (n1-1, n2-1). Here, n1=1236-108=1124 and n2=166-12-21=133. Or do you use n2=166-12=154? Since you report 153=154-1 as the number of degrees of freedom, it seems you do not subtract 21 "incomplete responses" (line 308) — why? In any case, my calculated values of T (-3.093383 if n2=133 or -3.309464 if n2=154) are different from yours (-3.130). Please check and rectify.
- Same for other instances of t-test usage — please check carefully.
- With regard to your methodology in Tables 1 and 2, please note that you are applying multiple tests on the same set of individuals, which warrants the need for a suitable correction of the significance. Please address this issue, for example by using the False Discovery Rate (FDR) techniques by Benjamini and Hochberg (1995).
- Lines 397–: Regression models are fitted rather formally, you need to demonstrate that linear regression is suitable by showing diagnostic plots such as residuals versus fitted values, etc.
Reviewer 2 Report
The objective of this study is to examine how life satisfaction has been changed during the Covid pandemic among fourth-grade elementary school students in the Seoul metropolitan area. The authors detect that the level of life satisfaction remained unchanged during pandemic.
Comments
- There is an important size difference between the sample pre-Covid and Covid. The sample size during COVID is 166 which is really small and it likely causes biased estimations and, hence, the results might be not valid. Perhaps, the authors might apply a difference-in-differences design to confirm whether the pandemic affected the life satisfaction level of these children (Huebener et al. 2021).
- Important control variables are not used: gender of children, school type (public or private), and socio-economic characteristics of the household (educational level of parents, income level of parents, family size, number of children….) which are potential influencers in the life satisfaction level. The authors should test whether the incorporation of these controls would change their results.
References:
Huebner, Mathias, Siegel, Nico, Spiess, C. Katharina, Wagner, Gert G. and Waights, Sevrin (2020) Parental well-being in times of Covid-19 in Germany. Rev Econ Household (2021) 19:91–122. https://doi.org/10.1007/s11150-020-09529-4
Reviewer 3 Report
Thank you for the opportunity to review this article. The work is interesting, but some aspects should be taken into account before publication.
The subject of the manuscript is very interesting and the time-consuming and intensive research and analysis should be appreciated.
Comments and suggestions for Authors:
Introduction:
- Introduction is quite long, I suggest you shorten the introduction a bit, focusing mainly on highlighting clearly the gap in the current knowledge. What is the novelty of the study?
- The beginning of the introduction, which is not a review of the literature, should be highly shortened.
Materials and Methods
- Please discuss power calculation and how the sample size is adequate.
- What were the group inclusion and exclusion criteria?
- Line 216 - Please specify the age range.
- There is information about obtaining the consent of the Bioethics Committee. Please provide the number of this consent.
Result:
- The appearance and readability of tables should be corrected.
Round 2
Reviewer 2 Report
No additional comments